# The Joint Influence of Tl^+^ and Thiol-Modifying Agents on Rat Liver Mitochondrial Parameters In Vitro

**DOI:** 10.3390/ijms23168964

**Published:** 2022-08-11

**Authors:** Sergey M. Korotkov, Artemy V. Novozhilov

**Affiliations:** Sechenov Institute of Evolutionary Physiology and Biochemistry, Russian Academy of Sciences, Thorez pr. 44, 194223 St. Petersburg, Russia

**Keywords:** Tl^+^, Ca^2+^, thiol-modifying agents, rat liver mitochondria, mitochondrial permeability transition pore

## Abstract

Recent data have shown that the mitochondrial permeability transition pore (MPTP) is the complex of the Ca^2+^-modified adenine nucleotide translocase (ANT) and the Ca^2+^-modified ATP synthase. We found in a previous study that ANT conformational changes may be involved in Tl^+^-induced MPTP opening in the inner membrane of Ca^2+^-loaded rat liver mitochondria. In this study, the effects of thiol-modifying agents (eosin-5-maleimide (EMA), fluorescein isothiocyanate (FITC), Cu(o-phenanthroline)_2_ (Cu(OP)_2_), and embelin (Emb)), and MPTP inhibitors (ADP, cyclosporine A (CsA), n-ethylmaleimide (NEM), and trifluoperazine (TFP)) on MPTP opening were tested simultaneously with increases in swelling, membrane potential (ΔΨ_mito_) decline, decreases in state 3, 4, and 3U_DNP_ (2,4-dinitrophenol-uncoupled) respiration, and changes in the inner membrane free thiol group content. The effects of these thiol-modifying agents on the studied mitochondrial characteristics were multidirectional and showed a clear dependence on their concentration. This research suggests that Tl^+^-induced MPTP opening in the inner membrane of calcium-loaded mitochondria may be caused by the interaction of used reagents (EMA, FITC, Emb, Cu(OP)_2_) with active groups of ANT, the mitochondrial phosphate carrier (PiC) and the mitochondrial respiratory chain complexes. This study provides further insight into the causes of thallium toxicity and may be useful in the development of new treatments for thallium poisoning.

## 1. Introduction

The pioneering research of Haworth and Hunter discovered a membrane transition in the inner membrane of Ca^2+^-loaded mitochondria [1]. This phenomenon is called the mitochondrial permeability transition pore (MPTP). In the earliest investigations, the main structural elements of the MPTP were considered to be the adenine nucleotide translocase (ANT), cyclophilin D (CyP-D), and voltage-dependent anion channels (VDAC) [2,3]. Subsequent studies of CyP-D and ANT-deficient mitochondria made it possible to consider phosphate symporter (PiC) and CyP-D as the MPTP’s main components, and to identify the ANT as its regulatory component [3,4,5,6,7]. In the subsequent research, the MPTP’s main components were found to add some subunits of Ca^2+^-modified F_1_F_0_-ATP synthase along with the ANT regulatory component [4,5,8,9]. According to the latest research, the MPTP’s main components are considered to be the complex of the Ca^2+^-modified ATP synthase and the Ca^2+^-modified ANT [4,6,10,11,12,13].

A significant mitochondrion calcium load induces MPTP opening in a high conductance state to permit molecules of size ≤ 1500 Da, which penetrate the inner mitochondrial membrane (IMM) [1,2]. If the calcium load is small, the pore can open in a low conductance state, and the membrane becomes permeable to inorganic ions (K^+^, Na^+^, Mg^2+^, Ca^2+^) and small molecules of size ≤ 300 Da [14]. An intermediate calcium load results in MPTP opening in a medium conductance state, which allows molecules of size 300–600 Da, and, in particular, sucrose, to pass through the inner membrane [15]. MPTP opening in the inner membrane is accompanied by high-amplitude mitochondrial swelling, the release of solutes (K^+^, Na^+^, Ca^2+^, Mg^2+^, Pi, adenine nucleotides) from the matrix, inner membrane potential decline and cytochrome *c* release in the intermembrane space [1,2,14,16]. It is necessary to take into account the interaction of some MPTP inducers with the IMM lipid component (acridines and oleate) [17,18]. MPTP opening has been found in calcium-loaded rat kidney mitochondria (RKM) in the presence of agaric acid, acridine orange and 10-N-nonyl acridine orange (probes for cardiolipin), and oleate (interacting with the inner membrane lipid component), as well as ethidium bromide (cationic probe) [17,18,19,20,21]. We showed previously that these phenomena resulted in Tl^+^-induced MPTP opening, which occurred in experiments in vitro with calcium-loaded mitochondria [22]. At the same time, Tl^+^ itself showed negligible interaction with mitochondrial thiol groups (Appendix A) [23]. However, this pore opening was noticeably increased in the presence of thiol reagents (phenylarsine oxide (PAO), 4,4′-diisothiocyanostilbene-2,2′-disulfonate (DIDS), mersalyl (MSL), high n-ethylmaleimide (NEM) and thiol oxidants (*tert*-butyl hydroperoxide (*t*BHP), diamide (Diam)) [24,25,26].

Eosin-5-maleimide (EMA), unlike the NEM, interacts more potently with ANT cysteine thiol groups, greatly sensitizes the MPTP to Ca^2+^, prevents the pore ADP from inhibiting and does not freely penetrate the inner mitochondrial membrane [3,27,28,29]. Experiments with submitochondrial particles (SMPs) showed the effects of EMA binding to the ANT dependents on free thiol group (Cys^159^, Cys^47^) activity and conformation of the ANT [3,27]. PAO or diamide pre-treatment modified the ANT Cys^159^ and also inhibited the binding of solubilized ANT to a glutathione S-transferase-CyP-D affinity column. These above effects were blocked by 100 μM EMA [3,27]. The EMA interaction with ANT and the ATP-dependent K^+^ channel was enhanced in the presence of CAT fixing ANT in c-conformation [30]. EMA potently inhibited beef heart mitochondrion swelling in a medium containing 120 mM NH_4_H_2_PO_4_ [31]. This effect resulted in EMA interacting with PiC-essential sulfhydryl groups placed on the cytoplasmic surface of the IMM [31].

Unlike EMA, fluorescein isothiocyanate (FITC) does not interact with ANT cysteines [24,26,32]. The electrogenic K^+^/H^+^ exchange across the IMM increased in experiments with energized rat liver mitochondria (RLM), and it was accompanied by mitochondrial swelling in a medium with 120 mM KCl and FITC [33]. FITC did not influence state 3U_FCCP_ (FCCP-uncoupled) respiration and respiratory control ratio (RCR_FCCP_) in RLM injected into a medium containing 120 mM KCl [34]. Further, FITC inhibited ADP transport into bovine heart mitochondria [32]. Inorganic phosphate (P_i_) uptake was inhibited completely with 200 µM FITC, and partly (50%) at 60 µM FITC [32]. Embeline (Emb) decreased the inner mitochondrial membrane potential (ΔΨ_mito_) and induced apoptosis, reactive oxygen species (ROS) production and cytochrome *c* release in human colon adenocarcinoma and MCF-7 breast cancer cells [35,36,37]. Conversely, FITC reacts with cysteine and the PiC lysine residues, easily penetrates the mitochondrial matrix, binds to α and γ subunits of the F1-ATP synthase and localizes in the inner membrane lipid component [32,33,34].

The reaction of copper complexes with *o*-phenanthroline (Cu(OP)_2_) with matrix-faced and external-faced vicinal SH groups results in MPTP opening in calcium-loaded mitochondria, due to the formation of sulfhydryl bridges in the inner membrane proteins, which increases the membrane ion permeability [38,39,40,41,42,43]. Cu(OP)_2_ can cross-link two ANT monomers in SMPs (RLM, rat heart mitochondria (RHM), bovine heart mitochondria (BHM)) via their Cys^57^ residues to produce a covalent dimmer; pre-treatment with PAO and Diam blocks dimmer formation [27,44]. These matrix-faced vicinal SH groups, unlike external-faced SH groups, were not available to react with PAO or Cu(OP)_2_ in the photo-modified RLM in the presence of 5 µM Ca^2+^ [38,39]. Cu(OP)_2_-induced morphological changes in mitochondria were in fact the same as those induced by Ca^2+^ [43]. All these changes disappeared in the presence of cyclosporine A (CsA), and the mitochondria structure was similar to the one found in Ca^2+^ free mitochondria [43]. Cu(OP)_2_ increased state 4 respiration in a Pi-free medium, which might have been due to the involvement of 29 kDa H^+^-ATPase protein [45].

Embelin (Emb, 2,5-dihydroxy-3-undecyl-1,4-benzoquinone) belongs to a class of ubiquinone analogs, and this reagent is a potent MPTP inducer [46]. Conversely, trifluoperazine (TFP) is a MPTP inhibitor that changes the pore voltage sensitivity and modulates adenine nucleotide binding through the surface charge effect in energized mitochondria [3]. However, it has previously been shown that the alkylation of glutathione and some Cys^57^ residues using 50 μM NEM reduced the pore calcium-sensitivity [3,27,47]. NEM increased this labeling in beef heart mitochondria, and Cu^2+^ did the same in rat kidney mitochondria, but it was completely inhibited by carboxyatractyloside (CAT) [30,48]. Ebselen (an antioxidant seleno compound) or ethidium bromide induced MPTP opening, which was accompanied by swelling and cytochrome *c* and Ca^2+^ release from calcium-loaded rat kidney cortex mitochondria, which was inhibited by CsA and NEM [21,49].

Currently, thallium industrial production and the use of these metal chemical compounds in various industries and medicine are increasing. At the same time, the industrial production of various synthetic and natural organic compounds is being intensified. Thus, the simultaneous intake of thallium and these compounds into the human body can enhance these effects of metal toxicity. On the other hand, the involvement of the inner membrane molecules with thiol and lysine residues in Tl^+^-induced MPTP opening has not been studied enough. Therefore, our aim in this study was to investigate the cooperative effects of thiol reagents (EMA, FITC, Cu(OP)_2_, and Emb) and MPTP inhibitors (ADP, CsA, low NEM, and TFP) on Tl^+^-induced MPTP opening in the inner membrane of calcium-loaded RLM. We studied the effects of these reagents on mitochondrial respiration in 4_0_ (basal), 4, 3 and 3U_DNP_ (DNP-uncoupled) states, the content of the inner membrane free SH groups, mitochondrial swelling, and ΔΨ_mito_ decline in experiments in vitro with calcium-loaded rat liver mitochondria.

## 2. Results

### 2.1. Effects of Tl^+^ and Thiol-Modifying Agents on the Swelling of Succinate-Energized Rat Liver Mitochondria

The thiol-modifying agents (EMA and FITC) induced only insignificant contractions of succinate-energized RLM, with the maximal effects found for the middle concentrations of 30 and 100 µM (Figure 1A,B; Appendix A). Conversely, Cu(OP)_2_ did not affect the swelling (Figure 1C), which markedly increased in the presence of NEM. The latter swelling was visibly diminished by the MPTP inhibitors in the series ADP + NEM < CsA + NEM < ADP + CsA + NEM (Figure 1C). Embelin of 50–100 μM somewhat accelerated the succinate-energized mitochondrion swelling (Figure 1D). EMA did not affect the swelling of non-energized mitochondria until 30 µM, regardless of the presence of ADP (short dash traces) in the medium A (Figure 2A; Appendix A). The swelling significantly decreased in experiments with 50 μM EMA both with and without ADP. However, the non-energized mitochondrion swelling was noticeably stimulated by an increase in FITC concentration from 200 to 300 μM (Figure 2B; Appendix A), and ADP (short dash traces) significantly hindered this effect. The contraction of succinate-energized RLM was slightly dependent on the concentration of EMA (Figure 2A), but it was notable in the presence of 100–300 μM FITC (Figure 2B; Appendix A). The contraction with appropriate concentrations of EMA or FITC was the same in the presence of ADP (Figure 2A,B).

The swelling of non-energized RLM decreased in the medium A containing Ca^2+^ and 5–10 μM EMA in comparison to the EMA-free experiments (Figure 3A; Appendix A). The swelling was replaced by mitochondrial contraction in similar experiments with 30–50 μM EMA (Figure 3A; Appendix A). The swelling of calcium-loaded mitochondria increased accordingly after the succinate injection into the medium with 5–10 μM EMA, in comparison to calcium-free experiments (Figure 3A; Appendix A). There was somewhat minimal swelling in the presence of 30 μM EMA after 7 min after adding mitochondria (Figure 3A). Before the succinate injection, the mitochondria showed additional increases in swelling in experiments with 30–300 μM FITC and Ca^2+^ (Figure 3B, short dash traces; Appendix A). The energized mitochondrion swelling increased with 30 μM FITC and Ca^2+^ (Appendix A). The experiments with 100 μM FITC (Figure 3B) were characterized by a transition from mitochondrial contraction (free of Ca^2+^) to swelling in the presence of succinate and Ca^2+^ in the medium A. The succinate-energized mitochondria contraction remained in similar experiments with 200–300 μM FITC and Ca^2+^, but it was less pronounced in comparison with that found in the experiments free of calcium. The swelling of non-energized mitochondria was enhanced from 15 to 100 μM TFP, and it was independent of the presence of calcium in the medium A (Figure 3C; Appendix A). The contraction of succinate-energized mitochondria consequentially weakened from 15 to 50 μM TFP. The contraction weakening was more pronounced in similar experiments with calcium-loaded mitochondria, and it was inhibited completely at 100 μM TFP, both with calcium and without calcium in the medium (Figure 3C).

The swelling of succinate-energized calcium-loaded RLM was accelerated by 5–10 μM EMA, with the minimal effect observed at 30 μM EMA (Figure 4A; Appendix A). ADP inhibited the swelling, and this effect was maximal in experiments with 30 μM EMA. The calcium-induced swelling of energized mitochondria was attenuated in increasing FITC concentrations from 30 to 200 μM, and this effect was more visible in the presence of ADP (Figure 4B; Appendix A). The 3 μM complex Cu(OP)_2_ slightly accelerated the swelling, which was inhibited by ADP (Figure 4C). It should be emphasized that Cu^2+^ (unlike Ca^2+^) forms very stable complexes with *o*-phenanthroline [50,51]. Thus, we can conclude that this effect was due to the Cu(OP)_2_ complex action, which affects mitochondria, and not due to Ca^2+^. Embelin of 25–100 μM (Figure 4D; Appendix A) slightly prevented the calcium-loaded mitochondrion swelling. The swelling ADP inhibition decreased in experiments with 25–50 μM Emb, and it was completely eliminated at 100 μM Emb (Figure 4D; Appendix A). The swelling of non-energized mitochondria correspondingly decreased in the presence of Ca^2+^, ADP and 10–50 μM EMA, with the maximal effect observed in experiments with 50 μM EMA (Figure 5A; Appendix A). The EMA-induced swelling of succinate-energized and calcium-loaded mitochondria was markedly inhibited by ADP, especially in the presence 50 μM EMA (Figure 5A; Appendix A). The FITC-induced swelling of non-energized mitochondria decreased in the presence of ADP, with the maximal effect found in experiments with 200 μM FITC (Figure 5B; Appendix A). The contraction of succinate-energized mitochondria in the medium with comparable FITC concentrations was markedly more pronounced in the presence of ADP (Figure 5B; Appendix A).

The MPTP inhibitors (ADP, CsA, NEM) visibly decreased the swelling of succinate-energized and calcium-loaded mitochondria (Figure 6; Appendix A) in experiments with the thiol-modifying agents (EMA, FITC, embelin, Cu(OP)_2_). The swelling in experiments with 10 μM EMA and 100 μM FITC accordingly decreased in the series: the agent alone > control > CsA, NEM > CsA + NEM > ADP + CsA, ADP + NEM > ADP (Figure 6A,B; Appendix A). Experiments with Emb were characterized by a similar series but CsA alone showed no effect on the swelling, with a weak influence of ADP and NEM observed (Figure 6C; Appendix A). The Cu(OP)_2_-induced swelling in the mitochondria (Figure 6D; Appendix A) was completely inhibited by ADP or CsA. However, the complex in the presence of NEM induced maximal mitochondrial swelling, which decreased in the series: Cu(OP)_2_ + NEM > Cu(OP)_2_ + NEM + CsA > Cu(OP)_2_ > control > Cu(OP)_2_ + NEM + ADP > Cu(OP)_2_ + NEM + ADP + CsA (Figure 6D; Appendix A). RLM preswollen in the medium A with Ca^2+^ (control) showed additional succinate-induced swelling (Figure 7; Appendix A), which was more extensive in the presence of 10 μM EMA or 100 μM FITC (panels A and B). The swelling was inhibited by ADP, CsA, and NEM (Figure 7A,B; Appendix A); however, the similar effect of Mg^2+^ was not as noticeable. Conversely, the mitochondrial contraction in the free calcium medium (Figure 7B, trace 0) was more pronounced in the presence of 100 μM FITC, as well as that of ADP, CsA, and NEM (not presented here). The succinate-induced swelling of calcium-loaded mitochondria remained the same both in the absence and in the presence of 3 μM Cu(OP)_2_ (Figure 7C,D). The succinate-energized mitochondria contracted in the free calcium medium with 3 μM Cu(OP)_2_ alone and swelled in the presence of 3 μM Cu(OP)_2_ and NEM (Appendix A). The swelling of succinate-energized RLM, preswollen in the medium with Ca^2+^, Cu(OP)_2_, and NEM (Figure 7D; Appendix A), was visibly inhibited by CsA with/without ADP but not by ADP alone. The swelling in experiments with 50 μM TFP decreased in the series: TFP + ADP > TFP alone > control, Mg^2+^ > NEM, ADP + NEM > CsA + NEM, CsA + NEM + ADP (Figure 7E; Appendix A).

### 2.2. Effects of Tl^+^ and Thiol-Modifying Agents on Respiration and ΔΨ_mito_ of Succinate-Energized Rat Liver Mitochondria

We showed previously [52] that state 3 and state 3U_DNP_ respiration diminished in increasing TlNO_3_ concentrations from 25 to 75 mM in a 400 mOsm medium with 125 mM KNO_3_. However, state 3U_DNP_ respiration was slightly reduced in a 260 mM medium with 75–125 mM TlNO_3_ and sucrose (Appendix A) [53]. On the other hand, we found that this respiration was markedly inhibited by 10–40 µM Cd^2+^ (Appendix A) [54]. Therefore, to evaluate the joint effect of Tl^+^ and thiol-modifying agents, we studied the respiration with RLM injected into the medium B containing 25 mM TlNO_3_ and 125 mM KNO_3_ (Figure 8; Appendix A). The respiration in 3 and 3U_DNP_ states after the consistent addition of ADP and DNP into the medium B was partly inhibited by 50 μM EMA in experiments with RLM energized by glutamate and malate (Figure 8A, bold traces). Similar experiments with succinate-energized mitochondria were characterized by minor decreases in state 3 and state 3U_DNP_ respiration at 30 μM and 50 μM EMA, respectively (Figure 8A). FITC of 100–300 μM visibly inhibited state 3 respiration and partly inhibited state 3U_DNP_ in RLM energized by glutamate and malate in the medium B (Figure 8B). However, state 3 and state 3U_DNP_ respiration was apparently inhibited by 100 μM FITC and completely inhibited by 300 μM FITC in similar experiments with succinate-energized mitochondria (Figure 8B). State 3 and state 3U_DNP_ respiration was partly inhibited by 50 μM TFP and completely by 100 μM TFP in experiments with RLM energized by glutamate with malate or succinate alone (Figure 8C). Similar results were found in experiments with succinate-energized mitochondria and 50–100 μM embelin (Figure 8D). State 4 respiration increased in the presence of 50–100 μM embelin (Figure 8D). DNP-uncoupled (3U_DNP_) respiration was visibly inhibited by 50 μM EMA in experiments with succinate-energized mitochondria injected into the medium A (Figure 9A; Appendix A). Further, 10–30 μM EMA slightly affected the respiration. State 3U_DNP_ respiration gradually decreased as FITC concentration increased from 100 to 300 μM (Figure 9B). On the other hand, 3 μM Cu(OP)_2_ did not influence the respiration, regardless of the presence of NEM in medium A (Figure 9C).

The decrease in state 3U_DNP_ respiration in calcium-loaded mitochondria was less pronounced in the presence of 10 μM EMA, and this effect was more noticeable if NEM alone or ADP with CsA were added into the medium A before calcium (Figure 10A; Appendix A). The decrease in calcium-induced respiration was even less pronounced in similar experiments with 30–50 μM EMA, with a maximum effect observed at 30 μM EMA (Figure 10A). Conversely, the decrease was more pronounced in similar experiments with 200–300 μM FITC compared both to experiments with 100 μM FITC (Figure 10B) and the control experiments with Ca^2+^ alone (Figure 10E), which were indistinguishable from each other. Cu(OP)_2_ prevented Ca^2+^-induced decreases in 3U_DNP_ respiration RLM, but this effect was eliminated if NEM was injected into the medium A before the complex (Figure 10C). The decrease induced by the joint presence of Cu(OP)_2_ and NEM was distinctly attenuated by MPTP inhibitors (ADP, CsA) in the series ADP < CsA < ADP + CsA (Figure 10C). However, 3 μM Cu^2+^ did not affect state 3U_DNP_ respiration regardless of the presence of Ca^2+^ and CsA (Figure 10D). This effect was inhibited by NEM and CsA (Figure 10D).

The fluorescent dye safranin O was used to assess the inner mitochondrial membrane potential (ΔΨ_mito_). The fluorescence decrease in the mitochondrial suspension after the addition of succinate into the medium C was the result of the dye uptake by energized mitochondria due to the appearance of the inner membrane potential. Figure 11 shows that the used agents (EMA, FITC, embelin) slightly reduced ΔΨ_mito_. Regardless of the presence of agents (EMA, FITC, Cu(OP)_2_, embelin), a noticeable decrease in the potential occurred after the addition of calcium into the medium (Figure 11). However, this effect was completely eliminated if this medium was supplemented by ADP and CsA.

### 2.3. The Joint Effects of Tl^+^ and Thiol-Modifying Agents on the SH Group Content in Rat Liver Mitochondria

Absorption measurement at 412 nm in buffer with DTNB makes it possible to estimate the free thiol group content in mitochondrial proteins [55,56]. We showed earlier that Tl^+^ did not influence the content in succinate-energized RLM injected into the medium A [23]. Figure 12A shows that the free thiol group content of mitochondrial proteins was slightly affected by 10–30 µM EMA in experiments with RLM injected into the medium A. Similar experiments showed some decreases in the thiol content in the presence of 100–150 μM FITC (Figure 12B) or 100 μM TFP (Figure 12D). Some increases in the content were observed in the presence of 3 μM Cu(OP)_2_ (Figure 12C). However, the thiol content changed little in the calcium-loaded mitochondria injected into the medium A containing these thiol-modifying agents (EMA, FITC, Cu(OP)_2_, FTP) (Figure 12A–D). The content declined in similar experiments in the medium containing these agents with ADP, CsA, and NEM (Figure 12A–C). Figure 13 shows that the content of tested free thiol groups rose in the same experiments in a medium containing 150 mM sucrose instead of 75 mM TlNO_3_, as well as these thiol-modifying agents (EMA, FITC, Cu(OP)_2_, TFP). The content did not change in similar experiments with calcium-loaded mitochondria injected into the latter medium (Figure 13A–D). Some decrease in the group content was found in similar experiments with calcium-loaded mitochondria in the presence of NEM (Figure 13A–C). Similar results were obtained by us in earlier experiments with this medium containing thiol reagents (PAO, *t*BHP, Diam, DIDS, MSL) [24,25,26].

## 3. Discussion

Effects of thiol-modifying agents (EMA, FITC, Cu(OP)_2_, TFP, embelin) were studied in vitro in experiments with rat liver mitochondria. These effects on Tl^+^-induced MPTP opening were multidirectional and depended on the agents’ concentration. The multidirectionality of the agents’ effects on RLM may have been due to their different reactions with mitochondrial thiol groups of the adenine nucleotide translocase, respiratory complexes, and the inner membrane. EMA and FITC showed weak interaction with mitochondrial thiol groups in calcium-free experiments. EMA, Cu(OP)_2_, and embelin can be attributed as weak MPTP inducers, while FITC and TFP showed some pore inhibition. This study provides further insight into the causes of thallium toxicity, and may be useful in the development of new treatments for thallium poisoning.

Cys^159^ is known to be located in the matrix adenine nucleotide-binding site of the ANT [27]. Due to various conformational changes, the availability of ANT SH groups for reagents either increases or, conversely, decreases depending on conformational changes in the ANT structure [28,49,57]. Experiments with isolated mitochondria and SMP showed that EMA (the aromatic anionic maleimide), larger than NEM, preferentially attacked Cys^159^ [3,27,28,49,57,58]. At the same time, the EMA affinity for the ANT Cys^59^ was not so great [3,28]. We previously found that thiol reagents (PAO, MSL, high NEM) and thiol oxidants (*t*BHP, Diam) markedly increased the energized mitochondrion swelling in a calcium-free medium with TlNO_3_ and KNO_3_ [24,25,26]. In contrast, in this study, EMA did not affect the energized mitochondrion swelling in similar experiments (Figure 1A). This result is obviously due to the fact that EMA, in contrast to NEM and the above thiol reagents, does not penetrate the inner mitochondrial membrane [31]. Earlier investigations suggest that the ADP-induced decrease in the binding of EMA with ANT can be the result of possible competition between ADP and EMA to bind with ANT [21,57]. The ADP’s ability to inhibit MPTP opening in de-energized bovine heart mitochondria was reduced in the presence of EMA [29]. This fact is in good agreement with a certain slowdown in energized mitochondrial contraction in the presence of EMA and ADP observed in this study (Figure 2A, dotted traces).

At the same time, the weak EMA effect on the non-energized mitochondrion swelling in the medium with TlNO_3_ and KNO_3_ (Figure 2A) suggests that this reagent did not affect the IMM passive ion permeability. Figure 8A and Figure 9A show that EMA inhibited state 3 respiration, but at the same time it moderately reduced state 3U_DNP_ respiration in experiments with RLM energized with substrates of the first (glutamate + malate) and second (succinate in the presence of rotenone) respiratory complexes. Most likely, this result was due to the inhibition of ANT with the EMA interaction in associating with the corresponding PiC cysteines at a simultaneous EMA weak reaction with the mitochondrial respiratory chain complexes.

FITC is known to bind to the outer and inner surfaces of the IMM and to localize mainly in hydrophobic regions of mitochondrial proteins [33,34]. Further, it has been found to increase the IMM’s potassium and proton permeability [33]. Perhaps for this reason, FITC, on the one hand, accelerated the swelling of deenergized mitochondria in the medium with TlNO_3_ and KNO_3_ (Figure 2B), and on the other hand, this reagent did not affect the subsequent contraction of preswollen mitochondria after the injection of succinate into the medium, due to the FITC’s weak effect on succinate dehydrogenase activity, as evidenced by its weak effect on 3U_DNP_ state respiration (Figure 9B). ADP action on the IMM’s outer side is known to reduce the membrane ion permeability [2,14]. FITC easily penetrates into the RLM matrix and binds to α and γ subunits of ATP synthase, with its activity being almost completely suppressed, but high succinate dehydrogenase activity is preserved [33]. State 3 respiration was strongly inhibited and state 3U_FCCP_ was only partially reduced in experiments with FITC-pre-treated mitochondria [33]. Figure 8B and Figure 9B show that FITC inhibited state 3 respiration in the presence of first and second respiratory complex substrates, and partly affected state 3U_DNP_ in the presence of these complexes’ substrates. This result may have been due to the FITC interaction with ANT and succinate dehydrogenase cysteines, while there was no FITC interaction with the first mitochondrial respiratory complex. We found a similar effect previously in similar experiments with DIDS [24]. In conclusion, it should be noted here that the lack of swelling and the even weak contraction of mitochondria in the experiments with EMA and FITC (Figure 1A,B) may have been due to the activity preservation of the second respiratory chain complex under these experimental conditions (Figure 9A,B), and the weak interaction these reagents had with the mitochondrial thiol groups (Figure 12A,B).

If the pore (MPTP) is in the open state and some inducer binds to Cys^159^, the EMA’s binding to the ANT Cys^159^ decreases, and vice versa; in the absence of such binding, there is a more active binding of EMA to ANT. This opening is accompanied by swelling and cytochrome *c* and Ca^2+^ release from these mitochondria, as well as ROS production increase and ΔΨ_mito_ decline. Ethidium bromide uptake of RKM was inhibited by Ca^2+^ and MSL (due to its reaction with SH groups); it was negligibly inhibited by Mg^2+^, but not by NEM, CAT, and P_i_ [21]. Similarly, ΔΨ_mito_ decline (a safranin output from the mitochondrial matrix into a medium) was found in our experiments with calcium-loaded RLM and mersalyl due to Tl^+^-induced MPTP opening in the inner membrane [26]. The ANT affinity for EMA may increase because of Ca^2+^-induced conformational changes, which exposes more ANT cysteine residues to interact with EMA [30,57]. The present study found that 5–10 µM EMA potentiated the swelling of succinate-energized and calcium-loaded rat liver mitochondria in comparison to the control (Figure 3A and Figure 4A; Appendix A). However, this swelling was inhibited by 30–50 μM EMA. A similar swelling decrease occurred in the presence of 25–50 µM TFP (Figure 3C), which is an MPTP inhibitor [3,59].

The interaction of both diamide and PAO with Cys^159^ has been found to greatly decrease ANT labelling with EMA [27,58]. The binding of EMA with ANT increased with Ca^2+^ rising from 5 to 40 μM, and the binding was decreased by ADP and blocked by CAT [31,57]. Blocking Cys^159^ by EMA or high NEM or ebselen (an antioxidant seleno compound) greatly decreased ADP inhibition of the MPTP due to blocking ADP binding to the ANT thiol groups [27,29,49,57]. These findings are in good agreement with the swelling ADP inhibition observed in our experiments with 5–30 μM EMA (Figure 4A and Figure 5A; Appendix A) and with the swelling increase with 500 μM NEM alone [24] in the medium with TlNO_3_ and KNO_3_. A similar effect was exerted by Emb, which hindered ADP’s ability to inhibit calcium-induced mitochondrial swelling in the medium with TlNO_3_ and KNO_3_ (Figure 4D). Therefore, some research assigns Emb to the class of ubiquinone analogs that induce MPTP opening [46]. Emb inhibited state 3U_CCCP_ respiration, which was RLM energized by 5 mM glutamate with 2.5 mM malate [46]. Emb decreased state 3 respiration and inhibited state 3U_FCCP_ respiration in cells A549 in a medium containing pyruvate and malate [60]. In our study, as with EMA (Figure 8A), Emb partially reduced state 3 and 3U_DNP_ respiration in succinate-energized mitochondria (Figure 8D), but it had little effect on energized mitochondrial swelling (Figure 1D). Thus, in our case, we can describe EMA and Emb as weak MPTP inducers.

FITC, unlike EMA, does not interact with ANT, but it selectively labels the 34-kDa protein (PiC) at Lys^185^, both in bovine heart mitochondria and submitochondrial particles on the cytosolic and matrix sides [32]. The swelling of energized calcium-loaded mitochondria was attenuated in experiments with 100–200 μM FITC (Figure 4B). This FITC effect was even more substantial in the presence of ADP, and the mitochondria showed contraction instead of swelling (Figure 4B and Figure 5B). Thus, in this case, we can describe the summation of the effects of FITC and ADP. However, it is impossible to describe the complete inhibition of MPTP in experiments with FITC, since 100 μM FITC, unlike 10 μM EMA, did not prevent calcium-induced decreases in state 3U_DNP_ respiration, which were even more pronounced in experiments with 200–300 μM FITC (Figure 10A,B). This difference in the effects of FITC and EMA is probably due to the fact that FITC, similarly to MSL, reduces PiC activity. On the other hand, EMA, FITC and TFP did not affect the content of free thiol groups in calcium-loaded RLM (Figure 12A–C).

Another MPTP inhibitor (TFP) has previously been shown to prevent the formation of protein aggregates at the inner membrane permeabilization, induced by Ca^2+^ plus *t*BHP [59]. TFP has been found to change the MPTP voltage sensitivity and modulate adenine nucleotide binding through the surface charge effect [3]. Further, the antioxidant activity of trifluoperazine might also be responsible for its inhibitory effect on the Ca^2+^-induced MPTP [61]. Thus, the partial inhibition of calcium-induced swelling, and the lack of any changes to mitochondrial respiration in 4_0_ and 3U_DNP_ states of RLM energized by first and second respiratory complex substrates (Figure 3C and Figure 8C) may have been associated with the antioxidant and surface charge effects of TFP.

The reaction of PAO (inhibited by DTT) or Cu(OP)_2_ with matrix-faced and external-faced vicinal SH groups has been found to result in MPTP opening in the inner membrane of RLM and RKM, with both followed by decreases in Ca^2+^ retention capacity, ΔΨ_mito_ decline, and increases in mitochondrial swelling [38,39,41]. We found that 3 μM Cu(OP)_2_ did not affect swelling (Figure 1C), ΔΨ_mito_ (Figure 11), and the respiration in 4_0_ (a basal) and 3U_DNP_ states (Figure 9C) in experiments with RLM injected into calcium-free medium containing TlNO_3_ and KNO_3_. Similar experiments with calcium-loaded mitochondria and 3 μM Cu(OP)_2_ found a slight increase in mitochondrial swelling (Figure 4C), while the Ca^2+^-induced ΔΨ_mito_ decline (Figure 11) remained the same as in the control (experiments with calcium alone). It should be emphasized that Cu(OP)_2_ prevented decreases in the 3U_DNP_ state respiration in calcium-loaded rat liver mitochondria (Figure 10C). Previously, we found a similar result with 50 μM NEM, which significantly inhibited MPTP opening in experiments with calcium-loaded RLM, injected into the medium with TlNO_3_ and KNO_3_ in the presence of thiol reagents (PAO, *t*BHP, Diam, MSL) [24]. Further, 3 μM Cu(OP)_2_ showed negligible effect on the content of free thiol groups, regardless of the presence of Ca^2+^ in the medium (Figure 12C). These results allow us to consider Cu(OP)_2_ a weak MPTP inducer in the medium with TlNO_3_ and KNO_3_ due to its possible binding to the ANT Cys^57^ [27,44].

Decreases in the ADP-induced binding of EMA with ANT can be the result of possible competition between ADP and EMA to bind with ANT [21,57]. The ANT affinity for EMA may increase because of Ca^2+^- or Cu^2+^-induced conformational changes, which would expose more ANT cysteine residues to interact with EMA [30,57]. We found the maximum inhibitory effect in experiments with EMA and ADP (Figure 6A, Figure 7A and Figure 10A). The swelling of energized mitochondria in the presence of 10 µM EMA and Ca^2+^ increased in the series ADP < ADP along with CsA or NEM < CsA or NEM < control (Ca^2+^ alone) < EMA alone (Figure 6A and Figure 7A; Appendix A). The inhibition of Ca^2+^-induced decreases in state 3U_DNP_ respiration showed a similar pattern to these MPTP inhibitors (Figure 10A). The observed additive effect of EMA and ADP on the MPTP inhibition may have been related to the interaction of these reagents with the ANT Cys^159^.

TFP prevented Ca^2+^ and Mg^2+^ efflux from the mitochondria, mitochondrial swelling, ROS production, and ΔΨ_mito_ decline in experiments with calcium-loaded mitochondria in the presence of Diam, *t*BHP or P_i_ [59,61,62,63,64]. MPTP inhibition resulted in the simultaneous presence of TFP, and CsA was synergistic due to the TFP surface potential effect [61,64]. Swelling was inhibited more visibly in the presence of CsA or Mg^2+^ than with other reagents (ADP, NEM, DTT, GSH) in experiments with calcium-loaded RLM and TFP [61]. Earlier, we found [24] that CsA but not ADP showed more weak inhibition of the Tl^+^-induced MPTP in calcium-loaded RLM in experiments with a TlNO_3_ and KNO_3_ medium containing thiol oxidants (*t*BHP, Diam) or the thiol reagent PAO. However, in the present study, the CsA inhibitory effect was stronger than ADP in similar experiments with TFP (Figure 7E), possibly due to the TFP’s effect on the membrane surface potential and TFP antioxidant activity [61]. CsA or TFP showed additive swelling inhibition in Ca^2+^-loaded RLM [65]. Similar swelling inhibition was found herein in experiments with TFP and CsA (Figure 7E).

The thiol oxidant Cu(OP)_2_ reaction with calcium-loaded RLM resulted in the MPTP opening, which was inhibited by CsA or DTT [38,40,42]. CsA inhibited more potently the state 4 increase in experiments with Ca^2+^ than in those with Cu(OP)_2_ [43]. The Cu(OP)_2_-induced state 4 increase was observed in experiments with RLM in the presence of P_i_ only, possibly due to the involvement in this process of 29 kDa protein in H^+^-ATPase [45]. We also found a negligible state 4 respiration increase in the medium with TlNO_3_ and KNO_3_ free of NEM (Figure 10C). The presence of Cu(OP)_2_ did not, in fact, affect the swelling of energized calcium-loaded RLM (Figure 7C). This swelling inhibition series (Figure 7C) in experiments with Cu(OP)_2_ and MPTP inhibitors (ADP, CsA, NEM) was almost the same as in similar experiments with EMA (Figure 6A and Figure 7A), FITC (Figure 6B and Figure 7B), and Emb (Figure 6C) with maximum ADP effects. On the other hand, the state 3U_DNP_ respiration decrease was markedly inhibited in experiments with calcium-loaded RLM in the presence of NEM alone or ADP with CsA in experiments with EMA and FITC (Figure 10A,B). We earlier found a similar inhibition in similar experiments with thiol reagents (PAO, *t*BHP, Diam, MSL) and Ca^2+^ in the medium with TlNO_3_ and KNO_3_ [24,25,26]. Thus, in this case, we cannot rule out the participation of the studied reagents (EMA, FITC, Emb, Cu(OP)_2_) in the interaction with ANT active groups during Tl^+^-induced MPTP opening in the inner membrane of calcium-loaded RLM.

A previous study found that n-ethylmaleimide (NEM) inhibited MPTP induced by thiol agents or dithiol oxidants, but potentiated Cu(OP)_2_-induced MPTP opening, which was blocked by DTT [40]. The swelling of succinate-energized RLM increased in series 2.5 µM Cu(OP)_2_ + DTT, Cu(OP)_2_ + 0.85 µM CsA, Cu(OP)_2_ + 25 µM NEM + 5 µM Ca^2+^ + DTT (no swelling) << Cu(OP)_2_ + NEM + < Cu(OP)_2_ + NEM + Ca^2+^ [40]. The swelling of succinate-energized and FCCP-uncoupled RLM increased in series Ca2+ alone < Cu(OP)_2_ + Ca^2+^ < Cu(OP)_2_ + NEM + Ca^2+^ [40]. Similar NEM inhibition of Tl^+^-induced MPTP opening was found in our experiments with thiol reagents (PAO, *t*BHP, Diam, DIDS, MSL) [24,25,26] as well as thiol-modifying agents (EMA, FITC) and the MPTP inducer embelin (Figure 6, Figure 7 and Figure 10). This research showed that NEM visibly increased mitochondrial swelling and a Ca^2+^-induced state 3U_DNP_ decrease due to Tl^+^-induced MPTP opening in the presence of Cu(OP)_2_ (Figure 1C, Figure 6D and Figure 10C). On the other hand, the Cu(OP)_2_-induced swelling of succinate-energized and calcium-loaded RLM was inhibited by CsA [38,42,43]. A Cu(OP)_2_-induced increase was found in state 4_0_ respiration of succinate-energized RLM [43,45]. This state 4_0_ increase was also inhibited by CsA, NEM, and ADP. We also found a similar state 4_0_ increase in a medium containing TlNO_3_, KNO_3_, NEM, Cu(OP)_2_ (Figure 10C). This enhancement of the Cu(OP)_2_ effects in the presence of 50 μM NEM (Figure 1C, Figure 6D and Figure 10C), but not their attenuation, is paradoxical at first glance, and has been observed in similar experiments with the above thiol reagents [38,42,43] (Figure 6, Figure 7 and Figure 10); this phenomenon may be due to the fact that both NEM and Cu(OP)_2_ bind to the same thiol groups of ANT, which may be Cys^57^ [27,44].

The MPTP inhibitors (ADP, CsA) markedly prevented the Ca^2+^-induced effects (increased swelling, ΔΨ_mito_ decline, decreased respiration in 4, 3, 3U_DNP_ states) in experiments with succinate-energized and calcium-loaded RLM injected in a medium containing TlNO_3_, KNO_3_, and the thiol-modifying agents (Figure 6, Figure 7, Figure 10 and Figure 11). Based on these results, an increase in the concentration of free thiol groups in the inner membrane fragments should be obtained under these experimental conditions in the presence of these MPTP inhibitors, ADP, and CsA. On the contrary, we found some decreases in the free thiol group content in experiments with Ca^2+^-loaded RLM and used reagents (EMA, FITC, Cu(OP)_2_) in the presence of NEM alone or ADP with CsA (Figure 12A–C). The content decrease, on the one hand, was due to the NEM reaction with thiol groups. On the other hand, the decrease in experiments with ADP and CsA (Figure 12) may have resulted from some change to the inner membrane conformation. In this case, other thiol groups not associated with the Tl^+^-induced pore opening can be available to react with DTNB [65,66]. At the same time, a noticeable increase in the concentration of free thiol groups was found in a similar medium with sucrose free of Tl^+^ (Figure 13). A possible reason for this content increase may be due to the formation of matrix protein aggregates with IMM fragments; this could lead to thiol group content increases when reacting with DTNB [65].

## 4. Materials and Methods

### 4.1. Animals and Ethics

Male Wistar rats (250–300 g) were kept at 20–23 °C under 12 h light/dark cycle with free access to water ad libitum and the standard rat diet. All treatment procedures of animals were performed according to the Animal Welfare Act and the Institute Guide for Care and Use of Laboratory Animals.

### 4.2. Chemicals

The analytical grade chemicals were sodium dodecyl sulfate (SDS), KNO_3_, TlNO_3_, sucrose, ethylenediaminetetraacetic acid as disodium salt (EDTA), Mg(NO_3_)_2_, Tris-PO_4_, CaCl_2_, CuSO_4_, and 2,4-dinitrophenol (DNP). The next chemicals were purchased from Sigma (St. Louis, MO, USA): oligomycin, dithiothreitol (DTT), rotenone, eosin-5-maleimide (EMA), fluorescein isothiocyanate (FITC), trifluoperazine (TFP), o-phenanthroline, 2,5-dihydroxy-3-undecyl-1,4-benzoquinone—embelin (Emb), 5,5-dithio-bis-nitrobenzoic acid (DTNB), cyclosporine A (CsA), ethylene glycol-bis(β-aminoethyl ether) N,N,N′,N′-tetraacetic acid (EGTA), Tris-OH, ADP, n-ethylmaleimide (NEM), safranin O, and succinate. A column filled with a KU-2–8 resin from Azot (Kemerovo, Russia) was used after refining sucrose (1 M solution) from cation traces.

### 4.3. Mitochondrial Isolation

The isolation of rat liver mitochondria was carried out according to [67] in a buffer containing 250 mM sucrose, 3 mM Tris-HCl (pH 7.3), and 0.5 mM EGTA; next, mitochondrial sediment was washed out twice by resuspension/centrifugation in a medium containing 250 mM sucrose and 3 mM Tris-HCl (pH 7.3), and the mitochondria were finally suspended in 1 mL of the latter buffer. The quality of mitochondrial preparations was assessed in medium with 100 mM KCl and 20 mM Tris-HCl (for more detail, see Appendix A). The mitochondrial protein content assay was carried out according to Bradford; this was within 50–60 mg/mL.

### 4.4. Swelling of Mitochondria

Mitochondrial swelling (Figure 1, Figure 2, Figure 3, Figure 4, Figure 5, Figure 6 and Figure 7, and also see the Appendix A) was tested as decreases in A_540_ at 20 °C using a SF-46 spectrophotometer (LOMO, St. Petersburg, Russia). Mitochondria (1.5 mg of protein/mL) were added into a 1-cm cuvette with 1.5 mL of 400 mOsm medium A, which contained 75 mM TlNO_3_, 125 mM KNO_3_, 5 mM Tris-NO_3_ (pH 7.3), 2 μM rotenone, and 1 μg/mL oligomycin. The 400 mOsm media were used under detecting experimental parameters (swelling, oxygen consumption rates, ΔΨ_mito_, protein thiol content) with the view of checking the consistency and comparability of the events in the different experimental protocols. EMA, FITC, TFP, Cu(OP)_2_, Cu^2+^, Emb, Ca^2+^, succinate, NEM, ADP, and CsA were injected into the medium before or after mitochondria (see legends of Figure 1, Figure 2, Figure 3, Figure 4, Figure 5, Figure 6, Figure 7, Figure 8, Figure 9, Figure 10, Figure 11, Figure 12 and Figure 13). Typical traces from one of three independent different mitochondrial preparations are presented in Figure 1, Figure 2, Figure 3, Figure 4, Figure 5, Figure 6, Figure 7, Figure 8, Figure 9 and Figure 10.

### 4.5. Oxygen Consumption Assay

Mitochondrial respiration (natom O/min/mg of protein) was tested polarographically using Expert-001 analyzer (Econix-Expert Ltd., Moscow, Russia) in a 1.5-mL closed thermostatic chamber with magnetic stirring at 26 °C. Mitochondria (1.5 mg of protein/mL) were added into the medium A (Figure 9 and Figure 10) or the medium B (Figure 8) containing 25 mM TlNO_3_, 100 mM sucrose, 125 mM KNO_3_, 5 mM Tris-NO_3_ (pH 7.3), 3 mM Mg(NO_3_)_2_, 3 mM Tris-P_i_, and 2 μM rotenone. These media were supplemented by 5 mM succinate, which was additionally injected into these media before mitochondria. ADP of 130 μM (Figure 8) and DNP of 30 μM (Figure 8, Figure 9 and Figure 10) were correspondingly administrated into the media after 2 min recording of state 4_0_ to induce state 3 and state 3U_DNP_ respiration. The respiratory control ratio (RCR_ADP_) was calculated as a ratio of state 3 to state 4 (Figure 8, and also see the Appendix A). The RCR_DNP_ was accordingly quantified as a ratio of state 3U_DNP_ to state 4 or that of state 3U_DNP_ to a basal state respiration (Figure 8, Figure 9 and Figure 10, and also see the Appendix A).

### 4.6. Mitochondrial Membrane Potential

The inner membrane potential (ΔΨ_mito_) generated by injection of 5 mM succinate into a medium was tested according to Waldmeier et al. [68] (Appendix A). The safranin O fluorescence intensity (arbitrary units) in the mitochondrial suspension was tested at 20 °C using the microplate reader (CLARIOstar^®^ *Plus*, BMG LABTECH, Ortenberg, Germany) at 485/590 nm wavelength (excitation/emission). The mitochondria (0.5 mg of protein/mL) were added into the medium C containing 20 mM TlNO_3_, 125 mM KNO_3_, 110 mM sucrose, 5 mM Tris-NO_3_ (pH 7.3), 1 mM Tris-P_i_, 2 μM rotenone, 3 μM safranin O, and 1 μg/mL of oligomycin. ADP, CsA, EMA, FITC, Cu(OP)_2_, and Emb were injected into the medium before mitochondria (see the Figure 11 legend). Then, 5 mM succinate, 75 μM Ca^2+^, and 30 μM DNP were added into the medium after mitochondria (see the Figure 11 legend). The inner membrane potential (ΔΨ_mito_) discovered by the changes in safranin O fluorescence after the succinate injection was correspondingly taken as 175 mV (100% fluorescence change) in control experiments free of thiol-modifying agents, ADP, CsA, and Ca^2+^ [24,68]. The other cases’ fluorescence values were calculated relative to this control. A parallel fourfold measurement for each individual 300 μL aliquot was made from three independent preparations.

### 4.7. Determination of Protein Thiol Content

The protein thiol content was measured using an Ellman reagent (Appendix A) [55,56]. The RLM were added into 20 °C medium containing 75 mM TlNO_3_ (Figure 12) or 150 mM sucrose (Figure 13), as well as 5 mM Tris-NO_3_ (pH 7.3), 125 mM KNO_3_, 100 μM Ca^2+^ (where indicated), 2 μM rotenone, 5 mM succinate, and 1 μg/mL of oligomycin (more detail see [23]). In order to separate the inner membrane proteins from the matrix proteins, the suspended mitochondrial material after 5 min of incubation was processed through three successive freeze-thawing procedures using the above medium. The material was centrifugated further within 2 min at 10,000 rpm in a Beckman Coulter Microfuge 22R Centrifuge. The resulting mitochondrial sediment was washed in medium with 125 mM KNO_3_ and 5 mM Tris-NO_3_ (pH 7.3), and further centrifugated at 10,000 rpm in the final stage. The final sediment was dissolved in 1 mL of medium, which contained 100 µM DTNB, 0.5 mM EDTA, 0.5 M Tris-HCl (pH 8.3), and 0.5% SDS [23]. The protein thiol content was detected at 412 nm, and DTT was used for calibration (see Appendix A).

### 4.8. Statistical Analysis

The statistical differences in results and corresponding *p*-values were evaluated using two population *t*-tests (Microcal Origin, Version 6.0, OriginLab Corporation, Northampton, MA, USA). These differences are presented as percentages of the average (*p* < 0.05) from one of three independent experiments (Figure 1, Figure 2, Figure 3, Figure 4, Figure 5, Figure 6, Figure 7, Figure 8, Figure 9, Figure 10, Figure 11, Figure 12 and Figure 13). More detailed statistical analysis is provided in the Appendix A.

## 5. Conclusions

The inhibition of MPTP opening depends on a reagent’s ability to permeate the inner membrane, so the EMA’s ability to inhibit this pore was not so pronounced in comparison to one of a low NEM. The research on the FITC’s effect on mitochondrial swelling in the medium with TlNO_3_ and KNO_3_ confirmed the ability of FITC to increase the inner membrane ion permeability due to its interaction with the membrane and ANT cysteines. This research on calcium-loaded mitochondria found that EMA, Cu(OP)_2_, and Emb can be classified as weak MPTP inducers, because their reaction with mitochondria depends on the translocase cysteines’ accessibility, which is influenced by the ANT conformation. The differences in the effects of EMA and FITC may have been due to the fact that FITC (similarly to MSL) interacts with the cytoplasm-directed PiC cysteines of the inner membrane and does not with the ANT cysteines. However, taking into account the totality of the obtained results, the Tl^+^-induced MPTP opening in the inner membrane of calcium-loaded mitochondria may have been due to the interaction of our research reagents (EMA, FITC, Emb, Cu(OP)_2_) with active groups of ANT, PiC and the mitochondrial respiratory chain complexes. Since, according to recent data, the MPTP’s main components are the complex of the Ca^2+^-modified ANT and the Ca^2+^-modified ATP synthase, future research could investigate the synthase’s involvement in the Tl^+^-induced MPTP opening in the inner mitochondrial membrane. This study provides further insight into the causes of thallium toxicity and may be useful in the development of new treatments for thallium poisoning.

## Figures and Tables

**Figure 1 ijms-23-08964-f001:**
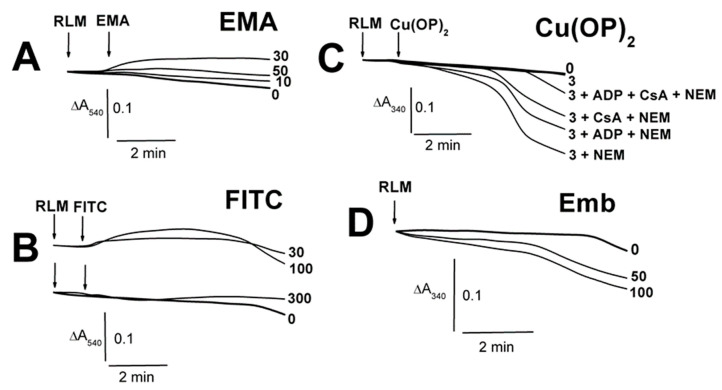
The influence of EMA, FITC, Emb, and Cu(OP)_2_ on the swelling of succinate-energized rat liver mitochondria. Mitochondria (1.5 mg of protein per mL) were added into the medium A containing 5 mM Tris-succinate (pH 7.3) and 75 mM TlNO_3_, as well as 50–100 μM Emb (**D**). Additions of mitochondria (RLM) and thiol-modifying agents (EMA, FITC, Cu(OP)_2_) are shown by arrows. The numbers on the right of the traces show concentrations (μM) of EMA (**A**), FITC (**B**), and Cu(OP)_2_ (**C**), as well as Emb (**D**), which was alone injected into the medium before mitochondria. In addition (**C**), 500 μM ADP (ADP), 1 μM CsA (CsA), and 50 μM NEM (NEM) were added into the medium before mitochondria (where indicated). For more details on the media composition in Figure 1, Figure 2, Figure 3, Figure 4, Figure 5, Figure 6 and Figure 7, see Section 4.

**Figure 2 ijms-23-08964-f002:**
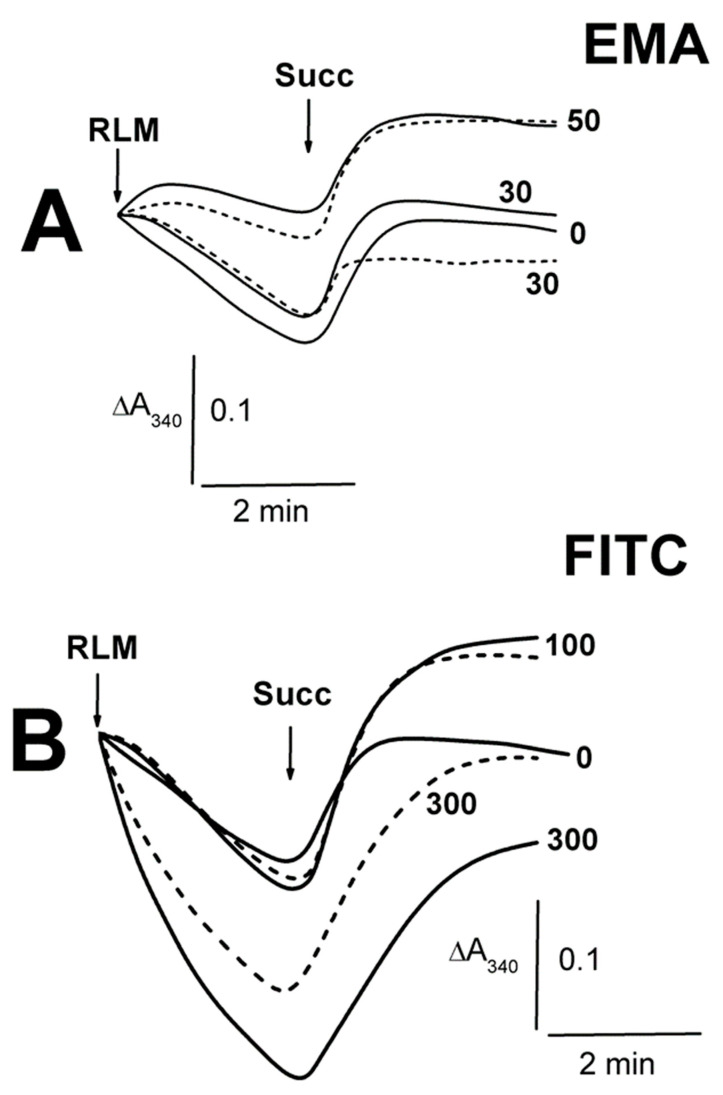
The influence of EMA and FITC on the Tl^+^-induced swelling of rat liver mitochondria. Mitochondria (1.5 mg/mL of protein) were placed into the medium A containing 75 mM TlNO_3_. The numbers on the right of the traces show concentrations (μM) of EMA (**A**) or FITC (**B**), injected into the medium before mitochondria. Additions of mitochondria (RLM) and 5 mM succinate (Succ) are shown by arrows. The medium was additionally supplemented by 500 μM ADP (short dash traces).

**Figure 3 ijms-23-08964-f003:**
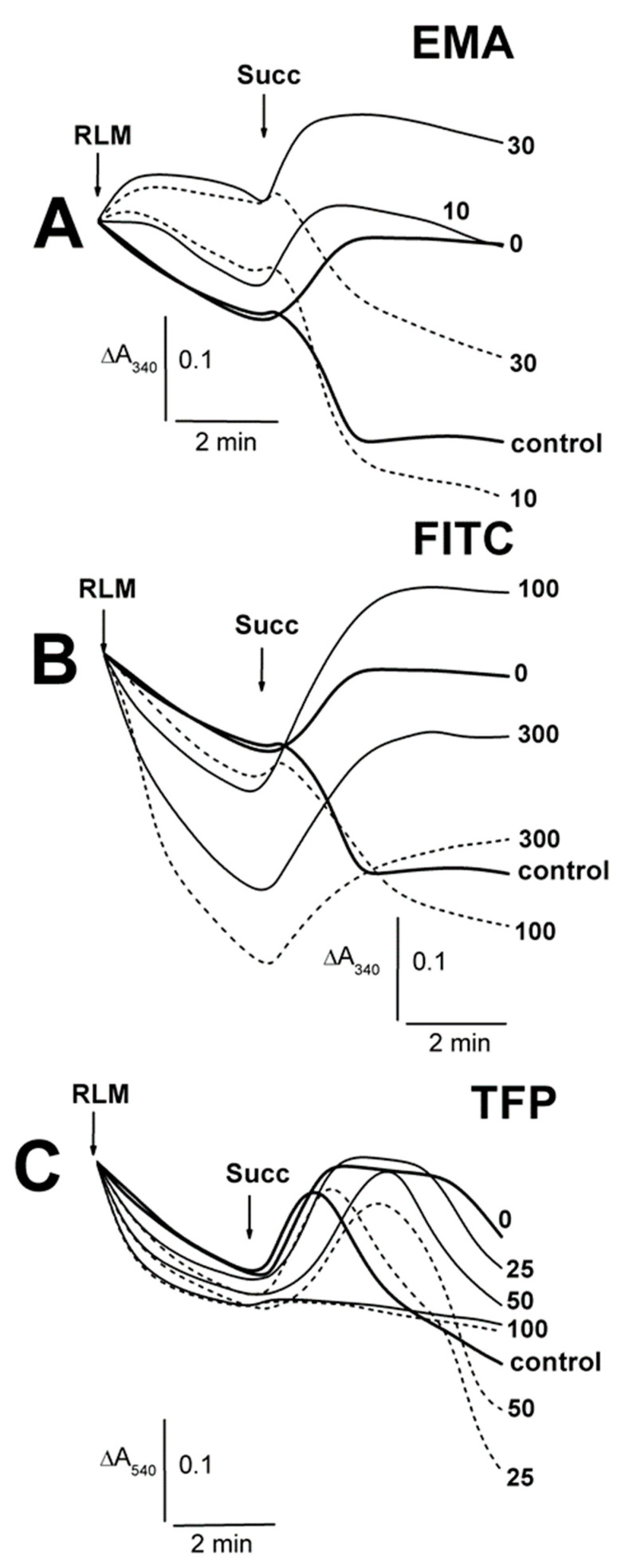
The influence of EMA, FITC and TFP on the Tl^+^-induced swelling of rat liver mitochondria. Mitochondria (1.5 mg/mL of protein) were injected into the medium A containing 75 mM TlNO_3_. The numbers on the right of the traces show concentrations (μM) of EMA (**A**), FITC (**B**), and TFP (**C**), which were injected into the medium before mitochondria. Additions of mitochondria (RLM) and 5 mM succinate (Succ) are shown by arrows. The medium was additionally supplemented by 50 μM Ca^2+^ (short dash traces). The bold traces show experiments free of Ca^2+^ (0) or ones with Ca^2+^ alone (control).

**Figure 4 ijms-23-08964-f004:**
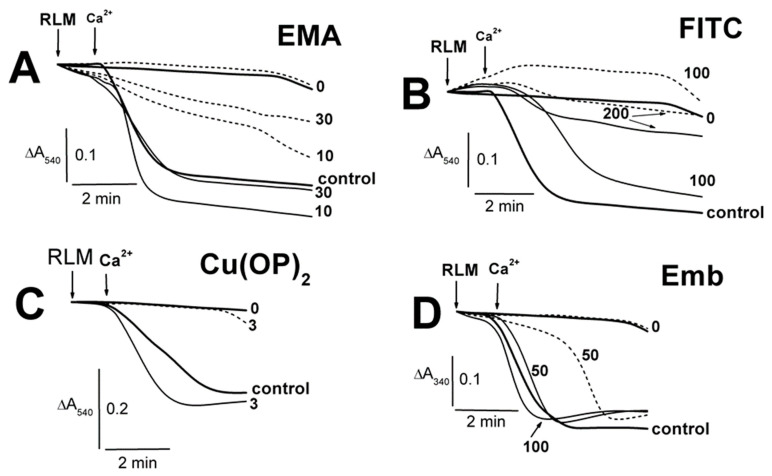
The influence of Ca^2+^ on the Tl^+^-induced swelling of succinate-energized rat liver mitochondria in the presence of EMA, FITC, Cu(OP)_2_, and Emb. Mitochondria (1.5 mg of protein per mL) were added into the medium A containing 75 mM TlNO_3_ and 5 mM Tris-succinate (pH 7.3). The medium was additionally supplemented by 500 μM ADP (short dash traces). The numbers on the right of the traces show concentrations (μM) of EMA (**A**), FITC (**B**), Cu(OP)_2_ (**C**), and Emb (**D**), which were injected into the medium before mitochondria. Additions of mitochondria (RLM) and 75 μM Ca^2+^ (Ca^2+^) are shown by arrows. The medium was additionally supplemented by 500 μM ADP (short dash traces). The bold traces show experiments free of Ca^2+^ (0) or ones with Ca^2+^ alone (control).

**Figure 5 ijms-23-08964-f005:**
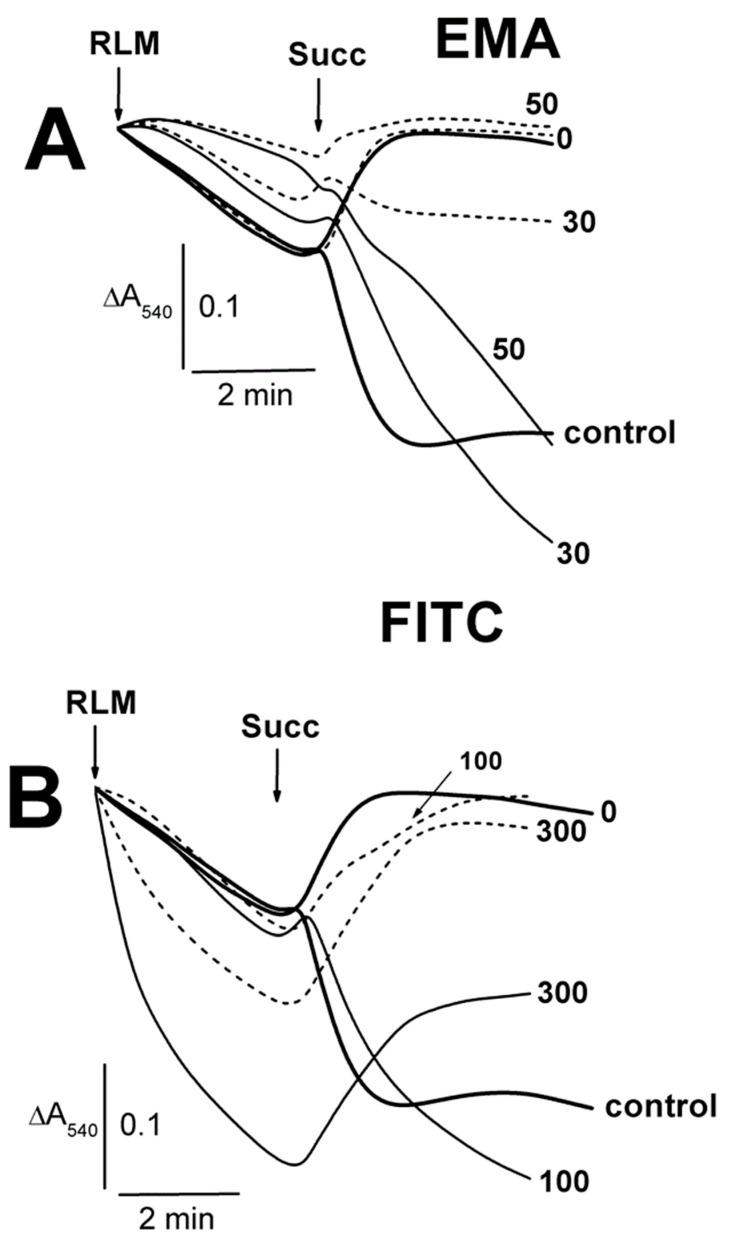
The influence of EMA and FITC on Tl^+^-induced swelling of rat liver mitochondria in the presence of Ca^2+^. Mitochondria (1.5 mg/mL of protein) were added into the medium A containing 75 mM TlNO_3_ and 75 μM Ca^2+^. The numbers on the right of the traces show concentrations (μM) of EMA (**A**) or FITC (**B**), injected into the medium before mitochondria. Additions of mitochondria (RLM) and 5 mM succinate (Succ) are shown by arrows. The medium was additionally supplemented by 500 μM ADP (short dash traces). The bold traces show experiments free of Ca^2+^ (0) or ones with Ca^2+^ alone (control).

**Figure 6 ijms-23-08964-f006:**
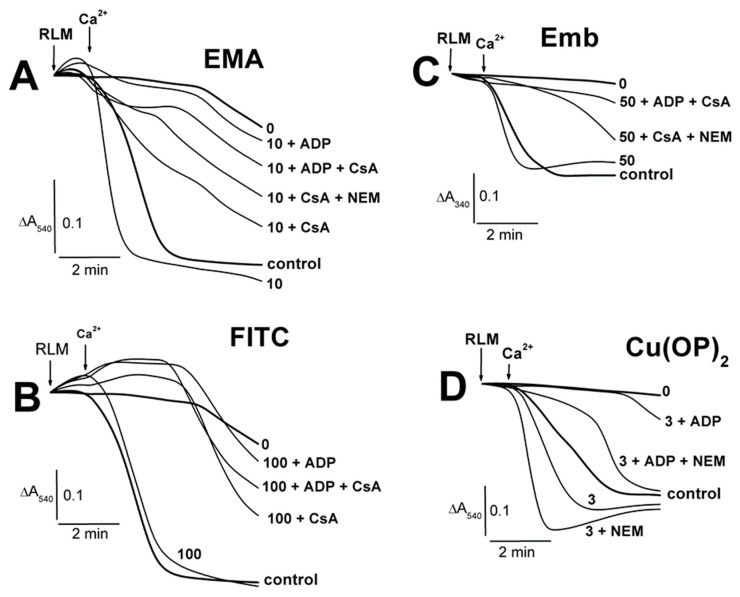
The influence of Ca^2+^ on the Tl^+^-induced swelling of succinate-energized rat liver mitochondria in the presence of EMA, FITC, Cu(OP)_2_, and Emb. Mitochondria (1.5 mg of protein per mL) were added into the medium A containing 75 mM TlNO_3_ and 5 mM Tris-succinate (pH 7.3). The numbers on the right of the traces show concentrations (μM) of EMA (**A**), FITC (**B**), Cu(OP)_2_ (**C**), and Emb (**D**), injected into the medium before mitochondria. Additions of mitochondria (RLM) and 75 μM Ca^2+^ (Ca^2+^) are shown by arrows. The medium (where indicated) was additionally supplemented by 500 μM ADP (ADP), 1 μM CsA (CsA), and 50 μM NEM (NEM). The bold traces show experiments free of Ca^2+^ (0) or ones with Ca^2+^ alone (control).

**Figure 7 ijms-23-08964-f007:**
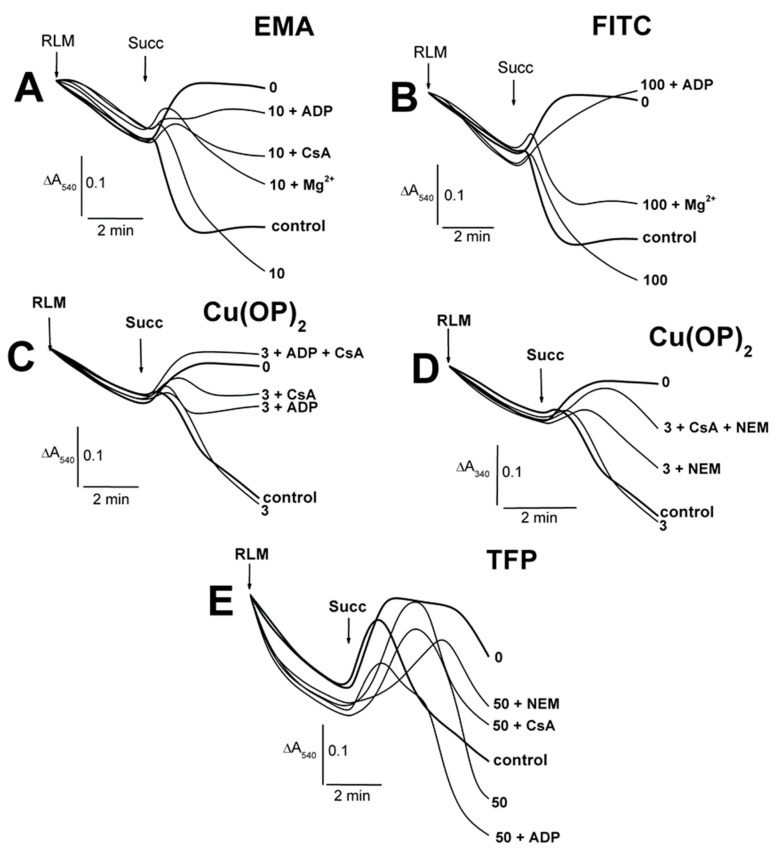
The influence of EMA, FITC, Cu(OP)_2_, and TFP on the Tl^+^-induced swelling of calcium-loaded rat liver mitochondria. Mitochondria (1.5 mg of protein per mL) were added into the medium A (pH 7.3) containing 75 mM TlNO_3_ and 75 μM Ca^2+^. The numbers on the right of the traces show concentrations (μM) of EMA (**A**), FITC (**B**), Cu(OP)_2_ (**C**,**D**), and TFP (**E**), which were injected into the medium before mitochondria. Additions of mitochondria (RLM) and 5 mM succinate (Succ) are shown by arrows. The medium (where indicated) was additionally supplemented by 500 μM ADP (ADP), 1 μM CsA (CsA), and 50 μM NEM (NEM). The bold traces show experiments free of Ca^2+^ (0) or ones with Ca^2+^ alone (control).

**Figure 8 ijms-23-08964-f008:**
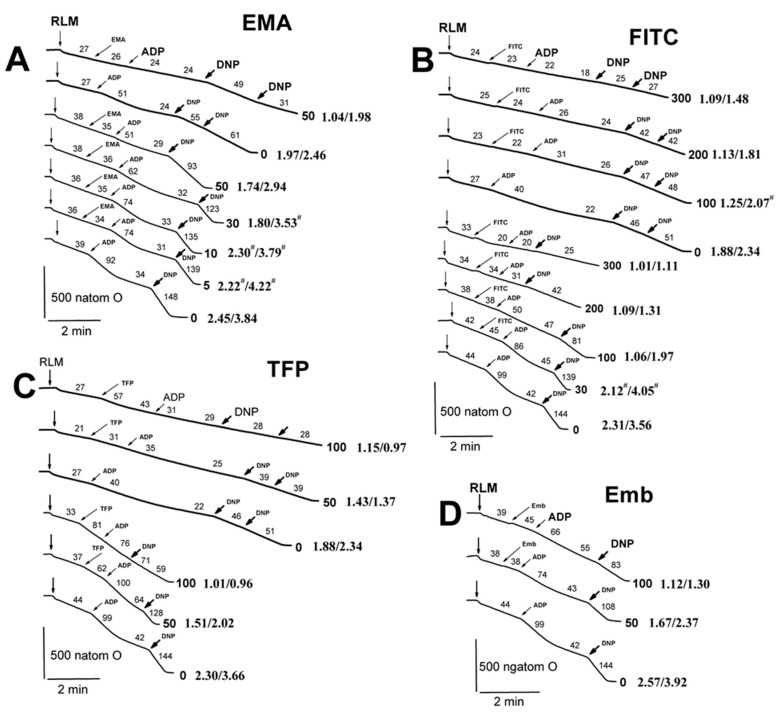
The influence of EMA, FITC, TFP, Emb and Ca^2+^ on oxygen consumption rates of rat liver mitochondria in the medium with 25 mM TlNO_3_. Mitochondria (1.5 mg/mL of protein) were injected into the medium B containing 5 mM succinate. Arrows show additions of mitochondria (RLM), 0–50 μM EMA (EMA), 0–300 μM FITC (FITC), 0–100 μM TFP (TFP), 0–100 μM embelin (Emb), 130 μM ADP (ADP), and double 15 μM DNP (DNP). Oxygen consumption rates (natom O min/mg of protein) are presented as numbers placed above experimental traces. The numbers in Arial and bold on the right of the traces show concentrations (μM) of EMA (**A**), FITC (**B**), TFP (**C**), and Emb (**D**). The numbers in Times New Roman and bold on the right of the traces show the ratios of the RCR_ADP_ and the RCR_DNP_ values (see the Section 4, and for more detail, see the Appendix A). Hash signs on the right of latter numbers indicate that differences between appropriate values of the RCR_ADP_ and the RCR_DNP_ are statistically insignificant to the values found in experiments free of EMA, FITC, TFP, and Emb.

**Figure 9 ijms-23-08964-f009:**
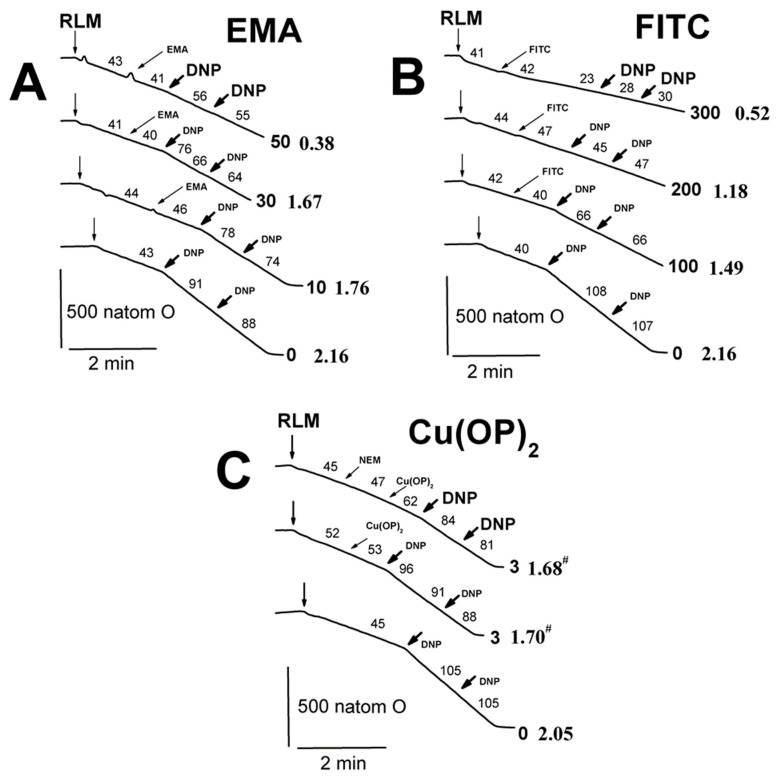
Influence of EMA, FITC, and Cu(OP)_2_ on the oxygen consumption rates of rat liver mitochondria in the medium with 75 mM TlNO_3_. Mitochondria (1.5 mg/mL of protein) were injected into the medium A containing 5 mM succinate. Arrows show additions of mitochondria (RLM), 0–50 μM EMA (EMA), 0–300 μM FITC (FITC), 0–3 μM Cu(OP)_2_ (Cu(OP)_2_), and double 15 μM DNP (DNP). The numbers on the right of the traces show concentrations (μM) of EMA (**A**), FITC (**B**), Cu(OP)_2_ (**C**). Oxygen consumption rates (natom O min/mg of protein) are presented as numbers placed above experimental traces. Numbers on the right of the traces in Times New Roman and bold show the RCR_DNP_ values (see the Section 4, and for more detail, see the Appendix A). Hash signs on the right of latter numbers indicate that differences between appropriate values of the RCR_DNP_ are statistically insignificant to the values found in experiments free of Cu(OP)_2_.

**Figure 10 ijms-23-08964-f010:**
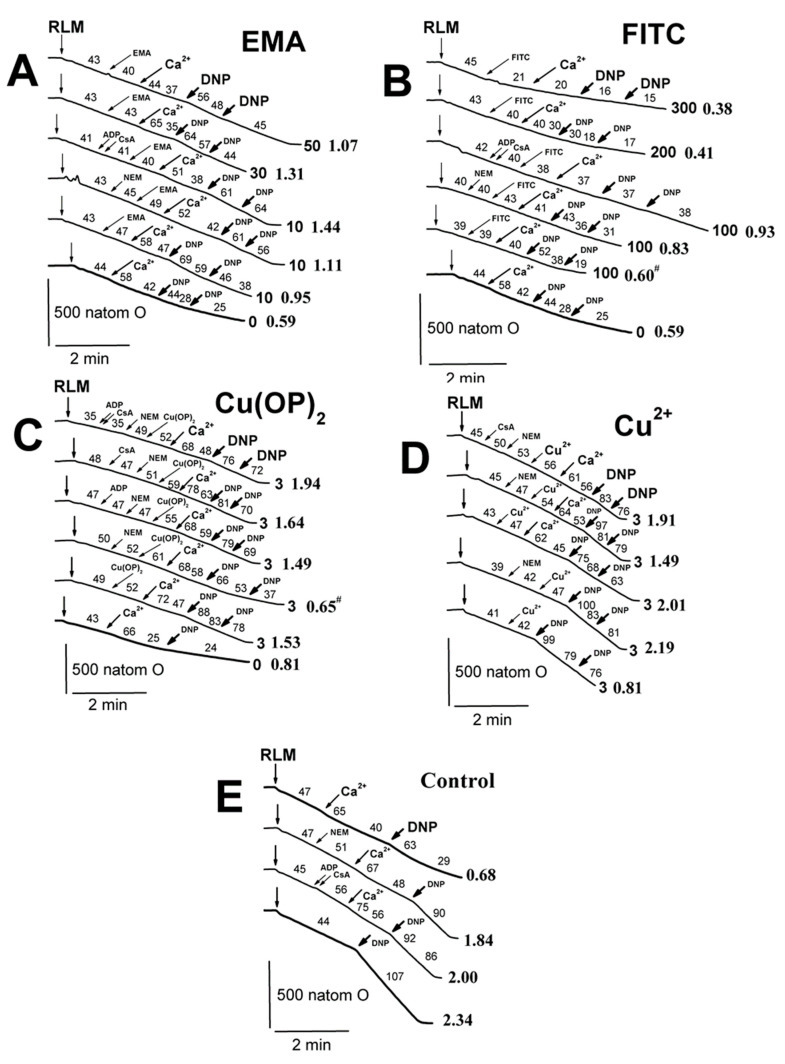
Influence of EMA, FITC, Cu(OP)_2_, Cu^2+^ and Ca^2+^ on the oxygen consumption rates of rat liver mitochondria in the medium with 75 mM TlNO_3_. Mitochondria (1.5 mg/mL of protein) were injected into the medium A containing 5 mM succinate. Arrows show additions of mitochondria (RLM), 0–50 μM EMA (EMA), 0–300 μM FITC (FITC), 0–3 μM Cu(OP)_2_ (Cu(OP)_2_), 3 μM Cu^2+^ (Cu^2+^), 75 μM Ca^2+^ (Ca^2+^), 500 μM ADP (ADP), 1 μM CsA (CsA), 50 μM NEM (NEM), and double 15 μM DNP in total (DNP). The numbers on the right of the traces show concentrations (μM) of EMA (**A**), FITC (**B**), Cu(OP)_2_ (**C**), Cu^2+^ (**D**), and free of thiol-modifying agents (**E**). Oxygen consumption rates (natom O min/mg of protein) are presented as numbers placed above experimental traces. Numbers on the right of the traces in Times New Roman and bold show the RCR_DNP_ values (see the Section 4, and for more detail, see the Appendix A). Hash signs on the right of latter numbers indicate that differences between appropriate values of the RCR_DNP_ are statistically insignificant to the values found in experiments free of EMA, FITC, Cu(OP)_2_, and Cu^2+^.

**Figure 11 ijms-23-08964-f011:**
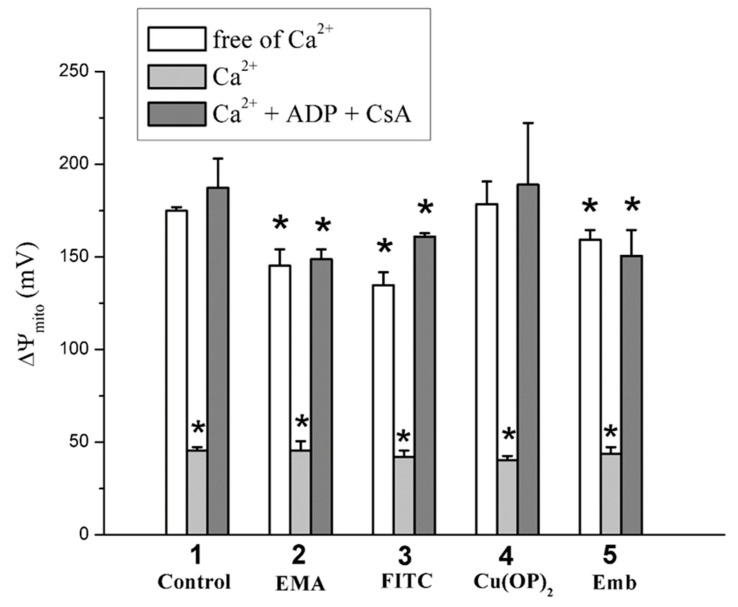
The effect of EMA, FITC, Cu(OP)_2_, Emb, and Ca^2+^ on the inner membrane potential (ΔΨ_mito_). Mitochondria (0.5 mg/mL of protein) were injected into the medium C containing 20 mM TlNO_3_. Ordinate shows ΔΨ_mito_ change (mV) calculated from safranine O fluorescence intensity (see Section 4). Numerals on the abscissa indicate used reagents: control experiments (1), 50 μM EMA (2), 100 μM FITC (3), 3 μM Cu(OP)_2_ (4), and 50 μM embelin (5), as well as 500 μM ADP and 1 μM CsA (where indicated) were correspondingly added into the medium C before mitochondria and Ca^2+^. Next, 75 μM Ca^2+^ was injected into the medium after mitochondria. Asterisks show significant differences from the control experiments free of Ca^2+^ and thiol-modifying agents (*p* < 0.05).

**Figure 12 ijms-23-08964-f012:**
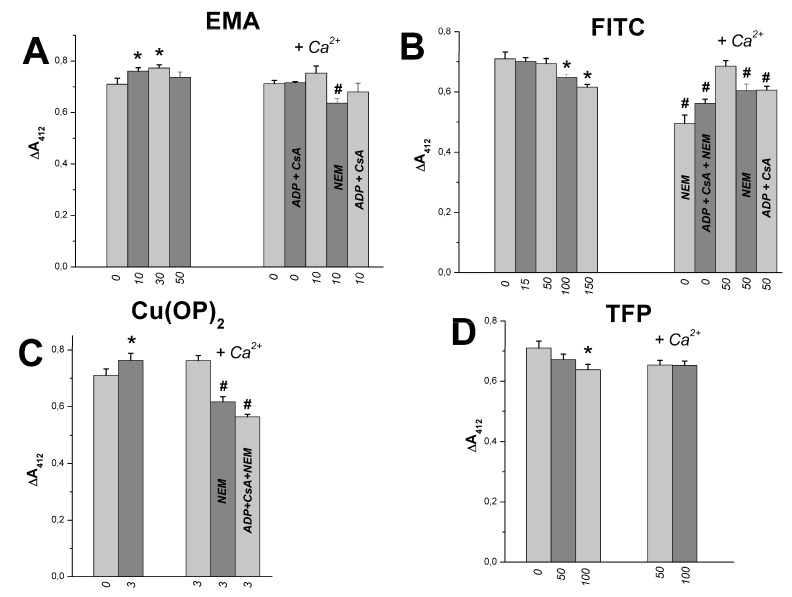
Effects of EMA, FITC, Cu(OP)_2_, TFP, and Ca^2+^ on the SH-group content in succinate-energized rat liver mitochondria in thallium nitrate medium. Mitochondria (1 mg protein/mL) were added into Eppendorf tube with 1 mL of the medium A (pH 7.3) containing 75 mM TlNO_3_, 125 mM KNO_3_, 5 mM succinate and 100 μM Ca^2+^ (Ca^2+^, where indicated). Reagents (EMA (**A**), FITC (**B**), Cu(OP)_2_ (**C**), TFP (**D**)) were injected into the medium before mitochondria. Then, mitochondria, after 5 min incubation at 20 °C, were sedimented at 10,000 rpm in the Beckman Centrifuge (see Methods). Numbers near the abscissa axis show the concentration (μM) of the reagents. Double measurements in the absorbance changes (ΔA_412_) were calculated for three different mitochondrial preparations, and these are shown as Means ± SEM. Asterisks and hash signs indicate correspondingly statistical differences between appropriate values and the control (free of thiol reagents or Ca^2+^ alone) which are statistically significant (*p* < 0.05).

**Figure 13 ijms-23-08964-f013:**
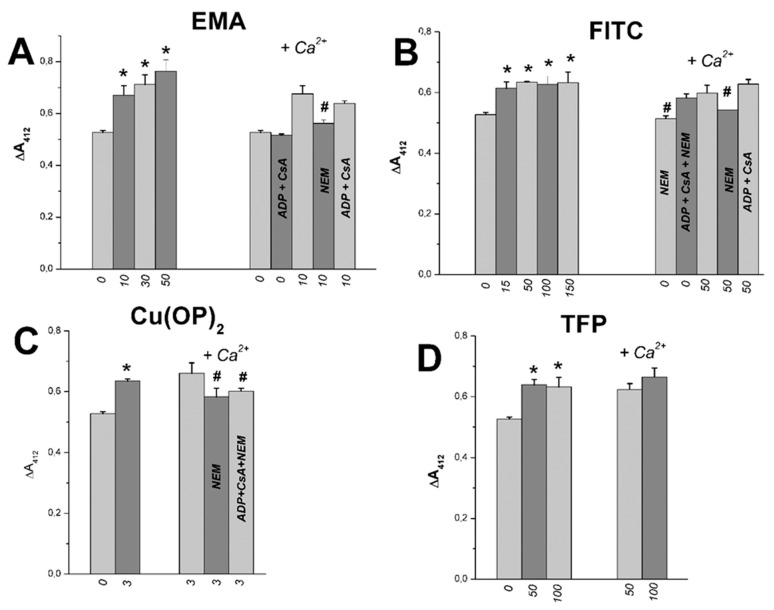
Effects of EMA, FITC, Cu(OP)_2_, TFP, and Ca^2+^ on the SH-group content in succinate-energized rat liver mitochondria in sucrose medium. Additions, experimental procedures and designations are as in Figure 12. The medium is the same as in Figure 12; however, 75 mM TlNO_3_ was replaced by 150 mM sucrose. Reagents (EMA (**A**), FITC (**B**), Cu(OP)_2_ (**C**), TFP (**D**)) were injected into the medium before mitochondria. Asterisks and hash signs indicate correspondingly statistical differences between appropriate values and the control (free of thiol reagents or Ca^2+^ alone) which are statistically significant (*p* < 0.05).

## Data Availability

The data that support the findings of this study are available from the corresponding authors, [S.M.K.], upon reasonable request.

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
