# Peer review of "The Joint Influence of Tl+ and Thiol-Modifying Agents on Rat Liver Mitochondrial Parameters In Vitro"

_ijms, 2022, doi:10.3390/ijms23168964_

Round 1

Reviewer 1 Report

This revised version of article described very interesting subjects. But, grammar and syntax errors are found throughout the manuscript. It seems the manuscript’s language has been corrected by a native English speaker available at our language center and appropriate corrections were replaced to the text. Also, Discussion is very long and should be revise. In my opinion, minor revision is necessary for correction of this article before acceptance of this article.

Author Response

My responses to the reviewer are in the attached files.

Reviewer 2 Report

This is an interesting article containing a big amount of experimental data. The paper should be published after a minor revision that should focus on following points:

1) The article should be accompanied by a short abstract in a form of highlights and/or a graphical abstract

2) The Authors should make order in size of the fonts they used in the text. Now, fonts of a different size are being used.

3) Compared to the "results" section, the introduction is too short. In particular, the Authors should explain precisely, why they decided to do their research studies and what is the significance of the results for applied toxicology.

4) The Authors should make a list of abbreviation of names of the compounds they used.

5) The long discussion should be supplied by a short summary and conclusion section.

6) Figures 2-7 contain too much information. This may be confusing especially for non-professional readers. I would suggest to show data for one-two concentrations only.  Other option is to replace the graph with traces by a graph with bars, separately for non-energised and energised mitochondria. Information about the concentration of Tl+ ions should be provided.

7) The Authors should explain the significance of the ratios (e.g. 1.04/1.98) on the right side of Figures 8 and 9.

8) I would suggest to replace the deltaF values in Figure 11 by the value of the inner membrane potential.

Author Response

Assigned Editor

Carlie Chen

IJMS Editorial Office

Dear Dr. Chen

There I present my replies to the two reviewer' comments on our manuscript "The joint influence of Tl+ and thiol-modifying agents on rat liver mitochondrial parameters in vitro". Despite a certain share of criticism, we are very grateful to these reviewers because of these remarks were very useful to improve our manuscript. Across new version our manuscript my corrections are highlighted in yellow. Please find our point by point response to each of the comments.

Sergey Korotkov, PhD in Biochemistry and MS in Chemistry

August 02, 2022

St. Petersburg

My Responses to Reviewer 2.

1.

The article should be accompanied by a short abstract in a form of highlights and/or a graphical abstract”.

These highlights are placed before the Discussion (page 10).

2.

The Authors should make order in size of the fonts they used in the text. Now, fonts of a different size are being used”.

This manuscript word document is made by 12pt Times New Roman.

3.

Compared to the "results" section, the introduction is too short”.

The introduction has been enlarged.

In particular, the Authors should explain precisely, why they decided to do their research studies…”.

We inserted the next text on Page 5 “Currently, thallium industrial production and the use of these metal chemical compounds in various industries and medicine are increasing. At the same time, the industrial production of various synthetic and natural organic compounds is being intensified. Thus, the simultaneous intake of thallium and these compounds into the human body can enhance these effects of metal toxicity”.

“ … and what is the significance of the results for applied toxicology”.

We inserted the next text on Page 18 (The Conclusion) “This study provides further insight into the causes of thallium toxicity and may be useful in the development of new treatments for thallium poisoning”.

4.

The Authors should make a list of abbreviation of names of the compounds they used”.

We inserted the list of abbreviation on Pages 1-2.

5.

The long discussion should be supplied by a short summary and conclusion section”.

Discussion of this manuscript has been reduced. The short summary (highlights) is placed on Pages 10-11. The conclusion section is on Pages 17-18.

6.

Figures 2-7 contain too much information. This may be confusing especially for non-professional readers. I would suggest to show data for one-two concentrations only.  Other option is to replace the graph with traces by a graph with bars, separately for non-energised and energised mitochondria. Information about the concentration of Tl+ ions should be provided”.

New Figures 2-7 show the information about one-two concentrations. The information about other used concentrations is indicated throughout the text in references to the supplement material. The information about the concentration of Tl+ ions is provided in the captions to these figures.

7.

The Authors should explain the significance of the ratios (e.g. 1.04/1.98) on the right side of Figures 8 and 9”.

The significance of the ratios (e.g. 1.04/1.98) on the right side of Figures 8 and 9 is clarified on in the captions to these figures “The numbers in Arial and bold on the right of the traces show concentrations of EMA, FITC, TFP, and Emb (mM). Numbers in Times New Roman and bold with slashes between them on the right of the traces show the ratios of the RCRADP and the RCRDNP values (see the Materials and methods, and for more detail, see the Supplementary data)”.

8.

I would suggest to replace the deltaF values in Figure 11 by the value of the inner membrane potential”.

The value of the inner membrane potential (deltaF, mV) is shown in Figure 11.